# Discovery of the exact 3D one-way wave equation

Kosmas L. Tsakmakidis [1] ✉ & Tomasz P. Stefański[2]

The standard wave equation describing symmetrical wave propagation in all directions in three dimensions, was discovered by the French scientist d'Alembert, more than 250 years ago. In the 20th century it became important to search for 'one-way' versions of this equation in three dimensions – i.e., an equation describing wave propagation in one direction for all angles, and forbiting it in the opposite direction – for a variety of applications in computational and topological physics. Here, by borrowing techniques from relativistic quantum field theory – in particular, from the Dirac equation –, and starting from Engquist and Majda's seminal, approximative one-way wave equations, we report the discovery of the exact one-way wave equation in three dimensions. Surprisingly, we find that this equation necessarily – similarly to the innate emergence of spin in the Dirac equation – has a topological nature, giving rise to strong, spin-orbit coupling and locking, and non-vanishing (integer) Chern numbers.

The well-known wave equation was first reported by d'Alembert in 1747[1], following critical insights by Bernoulli[2], Taylor[3], and Euler[4]. Since then, and particularly in the 20th century, with the emergence of computational techniques (requiring one-way absorbing boundary conditions) and topological physics[5], it became intriguing to identify, so called, 'one-way' wave equations in three dimensions – describing three-dimensional wave propagation in one direction, but completely forbiting it in the opposite direction. The breadth of those efforts can be measured by the literally thousands of papers that have been inspired by the first successful – yet approximative – effort by Engquist and Majda on arriving at such an 'one-way' wave equation, in 1977[6]. Even more recently, the search has again resurfaced following the rise of topological condensed matter[5], which involves robustly unidirectional waves – but whose one-way nature is usually ascertained from a dispersion band-diagram and/or underlying space- or time-symmetries, without explicit reference to an underlying *one-way wave* equation itself. In fact, until now the 'best' (least approximative) one-way 3D wave equation that we know of is still the one derived by Engquist and Majda more than 45 years ago, with various perturbative improvements[7-9]. Whereas until now we knew that topological waves are usually unidirectional, we are now led to the fundamental general

insight that the opposite too is true, namely that any wave propagating rigorously in a one-way manner for all angles, must inherently be topological in nature. Our work, establishing a foundational framework for the study of one-way waves in three dimensions, carries deep-ploughing consequences for the physics of topological and unidirectional wave transport, opening the road for a shift of emphasis from topology and symmetry directly to the nature of the underlying one-way transport, and can lead to new, simplified, designs of unidirectional and topological devices in physics and engineering[6].

We shall here report the discovery of just such an equation, starting our analysis from the *approximative* one-way wave[1-4] equations derived by Engquist and Majda in their pioneering work[5,7-9], and then, aided by the Dirac equation[6,10], arriving at *exact* one-way solutions of the wave equation in three dimensions. We shall then uncover in some detail the surprising topological nature that these new, exact solutions have. Whereas until now we knew that topological waves are usually unidirectional, we are now led to the fundamental general insight that the opposite too is true, namely that any wave propagating rigorously in a one-way manner for all angles, must inherently be topological in nature. Our work, establishing a foundational framework for the study of one-way

[1]Section of Condensed Matter Physics, Department of Physics, National and Kapodistrian University of Athens, Panepistimioupolis, GR - 157 84 Athens, Greece.
[2]Gdańsk University of Technology, Faculty of Electronics, Telecommunications and Informatics, ul. G. Narutowicza 11/12, 80-233 Gdańsk, Poland.
✉e-mail: ktsakmakidis@phys.uoa.gr

waves in three dimensions, carries deep-ploughing consequences for the physics of topological and unidirectional wave transport, opening the road for a shift of emphasis from topology and symmetry directly to the nature of the underlying one-way transport, and can lead to new, simplified, designs of unidirectional and topological devices in physics and engineering[6].

## Results

### The standard and approximative one-way wave equations

In three dimensions, the standard wave equation is given by:

$$\frac{\partial^2 U}{\partial x^2} + \frac{\partial^2 U}{\partial y^2} + \frac{\partial^2 U}{\partial z^2} - \frac{1}{c^2}\frac{\partial^2 U}{\partial t^2} = 0 \tag{1}$$

and we may define the operator $L = \frac{\partial^2}{\partial x^2} + \frac{\partial^2}{\partial y^2} + \frac{\partial^2}{\partial z^2} - \frac{1}{c^2}\frac{\partial^2}{\partial t^2} = L_x^2 + L_y^2 + L_z^2 - \frac{1}{c^2}L_t^2$, suggesting that Eq. (1) takes the form: $LU = 0$. Engquist and Majda proceeded[5] by breaking the operator L into two operators $L^+$ and $L^-$, such that $LU = L^+L^-U = 0$, with $L^+$ and $L^-$ being defined, in three dimensions, as:

$$L^+ = L_x + \frac{L_t}{c}\sqrt{1-\Pi^2} \tag{2}$$

and

$$L^- = L_x - \frac{L_t}{c}\sqrt{1-\Pi^2} \tag{3}$$

with $\Pi = \sqrt{(cL_y/L_t)^2 + (cL_z/L_t)^2}$. Crucially, in ref. 5, it is shown that the operation $L^-U = 0$ results *exactly* in a wave propagating in the *negative-x* direction (towards $x = 0$) only, *for all angles of incidence* – and similarly for $L^+U = 0$, in the *positive-x* direction (see Fig. 1). The approximation, here, arises from the way in which the square root in Eqs. (2), (3) is estimated: In[5], $U$ is assumed to be a scalar field, thus, if e.g. a second-order approximation is invoked, where $\sqrt{1-\Pi^2} \approx 1 - \Pi^2/2 + 0[\Pi^4]$, we have: $L^- \approx L_x - (L_t/c)(1-\Pi^2/2) = L_x - (L_t/c) + cL_y^2/(2L_t) + cL_z^2/(2L_t)$. From this last approximative expression for $L^-$, we arrive, using $L^-U = 0$, at the following (rather unfamiliar) *approximative* one-way wave equation:

$$\frac{\partial^2 U}{\partial x \partial t} - \frac{1}{c^2}\frac{\partial^2 U}{\partial t^2} + \frac{c}{2}\frac{\partial^2 U}{\partial y^2} + \frac{c}{2}\frac{\partial^2 U}{\partial z^2} = 0 \tag{4}$$

describing a wave propagating *solely in the negative-x direction* – but for a narrow range of incident angles, owing to $\Pi$ having been assumed 'small' ('small' values of $L_y$ and $L_z$ per $L_t$). Similar approximative one-way wave equations can be derived for all other remaining five directions ($+x$, $\pm y$, and $\pm z$). Hence, the innate approximation in Engquist and Majda's approach is the one concerning the square root of $1 - \Pi^2$ in Eqs. (2) and (3): Higher-order terms allow for progressively larger $L_y/L_t$ and $L_z/L_t$ terms, i.e. the approximate one-way equation is valid for a broader range of angles, whereas the zero-order approximation is valid only for $L_y = L_z = 0$, that is, for a one-dimensional transport only, leading to the familiar 1D one-way wave equation $\partial U/\partial x \pm (1/c)\partial U/\partial t = 0$.

### The exact one-way wave equation in three dimensions

To arrive at exact expressions for Eqs. (2) and (3), it should prove useful, from a pedagogical perspective, to be reminded of Dirac's insight for 'taking the square root' in a mathematically similar scenario in the relativistic theory of the electron[5,10]. Using the equation $E^2 = c^2p^2 + m^2c^4$ for relativistic massive particles, and making the assignments $E = \hbar\omega \leftrightarrow i\hbar(\partial/\partial t)$ and $\vec{p} = \hbar\boldsymbol{k} \leftrightarrow -i\hbar\vec{\nabla}$ for an assumed $e^{i(\mathbf{kr}-\omega t)}$ dependence, we obtain the Klein-Gordon equation: $-\hbar^2(\partial^2\psi/\partial t^2) = (-\hbar^2c^2\nabla^2 + m^2c^4)\psi$. At this point, Dirac's idea was to consider $\psi$ as not necessarily a scalar field, but a spinor field, and from Fig. 2 we immediately surmise that (taking $c = 1$) one may write, with no approximation(s) at all:

$$\sqrt{(p_x^2 + p_x^2 + m^2)I} = p_x\sigma_x + p_y\sigma_y + m\sigma_z \tag{5}$$

where $\sigma_x = \begin{pmatrix} 0 & 1 \\ 1 & 0 \end{pmatrix}, \sigma_y = \begin{pmatrix} 0 & -i \\ i & 0 \end{pmatrix}$, and $\sigma_z = \begin{pmatrix} 1 & 0 \\ 0 & -1 \end{pmatrix}$ are the Pauli spin matrices, and $I$ the unit matrix. We, thus, arrive at the exact, Dirac equation:

$$ih\frac{\partial\psi}{\partial t} = \left[-ih\left(\sigma_x\frac{\partial}{\partial x} + \sigma_y\frac{\partial}{\partial y}\right) + \sigma_z m\right]\psi, \text{ with } \psi = \begin{pmatrix} \psi_A \\ \psi_B \end{pmatrix} \tag{6}$$

which is consistent with the requirements of Lorentz covariance and respects particle conservation[5,10].

In a similar vein, assuming an $e^{-i(\mathbf{kr}-\omega t)}$ dependence, and with $c = 1$, we see from Eq. (3) that $L^- = L_x - i\sqrt{L_y^2 + L_z^2 - L_t^2}$, and thus, with the aid of Fig. 2, we may readily calculate – exactly, without any approximation(s) or fractional derivatives – the square root: $\sqrt{L_y^2 + L_z^2 - L_t^2} = \sigma_x L_y + \sigma_y L_z + i\sigma_z L_t$. As a corollary, from $L^-U = 0$, with

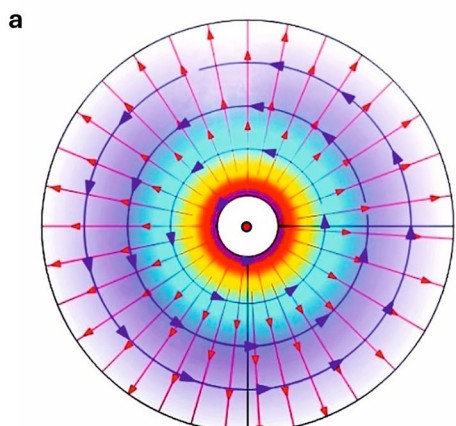

**Fig. 1 | The standard and the one-way wave equations. a** The standard wave equation describes a wave propagating symmetrically in all directions (thin red arrows) in three dimensions. Here, the blue arrows indicate a possible direction of the transverse magnetic field, for the case of an electromagnetic wave. **b** A one-way wave would literary be 'half' of the wave shown in (**a**), propagating for all transverse angles in only one direction, taken to be the 'positive' one, but not in the 'negative' direction.

$$(p_x^2 + p_y^2 + m^2)\begin{pmatrix} 1 & 0 \\ 0 & 1 \end{pmatrix} = \begin{pmatrix} m & p_x - ip_y \\ p_x + ip_y & -m \end{pmatrix}\begin{pmatrix} m & p_x - ip_y \\ p_x + ip_y & -m \end{pmatrix}$$

$$\sqrt{(p_x^2 + p_y^2 + m^2)}\, I = p_x \sigma_x + p_y \sigma_y + m\sigma_z$$

$$\sigma_x = \begin{pmatrix} 0 & 1 \\ 1 & 0 \end{pmatrix} \;;\; \sigma_y = \begin{pmatrix} 0 & -i \\ i & 0 \end{pmatrix} \;;\; \sigma_z = \begin{pmatrix} 1 & 0 \\ 0 & -1 \end{pmatrix}$$

**Fig. 2 | Dirac's insight for the relativistic equation describing an electron[5].** The idea is to take the square root of the operator $p_x^2 + p_x^2 + m^2$ (see also main text) 'without' taking the square root in the usual way. This calls for deploying matrices rather than scalar quantities. As shown in the figure, the three Pauli matrices immediately pop up, explaining naturally the existence of spin in an electron.

the use of the standard energy and momentum associations mentioned above, and the involutory properties of the Pauli matrices, namely $\sigma_x^2 = \sigma_y^2 = \sigma_z^2 = -i\sigma_x\sigma_y\sigma_z = I$, we may calculate the Hamiltonian for this case as:

$$H = \sigma_y p_y - \sigma_x p_z + \sigma_z p_x \tag{7}$$

i.e., the sought-after, **exact** (for *all* angles of incidence on the $x = 0$ plane) one-way wave (Weyl-like) equation is:

$$i\hbar \frac{\partial \psi}{\partial t} = (\sigma_y p_y - \sigma_x p_z + \sigma_z p_x)\psi = [(\boldsymbol{\sigma} \cdot \mathbf{p})|_y + (\boldsymbol{\sigma} \times \mathbf{p})|_y]\psi = [R_{ij}^{(2)}\sigma^i p^i + \varepsilon_{2jk}\sigma^j p^k]\psi \tag{8}$$

where $\psi$ is now a spinor field – not a scalar one, as in ref. [5] –, $\boldsymbol{\sigma}$ is the Pauli vector, the metric $R_{ij}^{(2)} = \mathrm{diag}(0, 1, 0)$, and $\varepsilon_{2jk}$ is the Levi-Civita symbol.

Further, for the assumed $e^{-itH/\hbar}$ time-dependence, and using well-known identities for matrix exponentials, the matrix governing the evolution of the left-going (towards $x = 0$) wave described by Eq. (8), turns out (in the wavevector basis) to be the following:

$$M = \begin{bmatrix} \cos(\omega t) - i\frac{k_x}{k}\sin(\omega t) & \left(-\frac{k_y}{k} + i\frac{k_z}{k}\right)\sin(\omega t) \\ \left(\frac{k_y}{k} + i\frac{k_z}{k}\right)\sin(\omega t) & \cos(\omega t) + i\frac{k_x}{k}\sin(\omega t) \end{bmatrix} \tag{9}$$

with eigenwaves (i.e., eigenvectors):

$$\psi_1 = \begin{bmatrix} -\frac{k_y}{k} + i\frac{k_z}{k} \\ i\left(1 + \frac{k_x}{k}\right) \end{bmatrix} \text{ and } \psi_2 = \begin{bmatrix} -\frac{k_y}{k} + i\frac{k_z}{k} \\ i\left(-1 + \frac{k_x}{k}\right) \end{bmatrix} \tag{10}$$

of square magnitudes $2\left(1 + \frac{k_x}{k}\right)$ and $2\left(1 - \frac{k_x}{k}\right)$, respectively, and eigenvalues $\lambda_{1,2} = \cos(\omega t) \pm i\sin(\omega t)$. From these expressions, we may calculate the canonical (orbital) momentum density[11], $\mathbf{p}^o = \mathrm{Re}\{\psi^\dagger \hat{\mathbf{p}}\psi\}$, for each eigenwave, arriving at:

$$\mathbf{p}_1^o = 2\hbar\mathbf{k}\left(1 + \frac{k_x}{k}\right) \tag{11a}$$

and

$$\mathbf{p}_2^o = 2\hbar\mathbf{k}\left(1 - \frac{k_x}{k}\right) \tag{11b}$$

where $k = \sqrt{k_x^2 + k_y^2 + k_z^2}$, while the $x$-, $y$-, $z$-components of the spin matrix $\mathbf{s} = \psi^\dagger \hat{\mathbf{S}}\psi$ (where $\hat{\mathbf{S}}$ is the standard matrix spin operator) for each

left-going eigenwave are:

$$\mathbf{s}_1 = \begin{bmatrix} s_{1,x} \\ s_{1,y} \\ s_{1,z} \end{bmatrix} = \begin{bmatrix} 2\frac{k_z}{k}\left(1 + \frac{k_x}{k}\right) \\ -2\frac{k_y}{k}\left(1 + \frac{k_x}{k}\right) \\ -2\frac{k_x}{k}\left(1 + \frac{k_x}{k}\right) \end{bmatrix} = \begin{bmatrix} \frac{p_{1,z}^o}{\hbar k} \\ -\frac{p_{1,y}^o}{\hbar k} \\ -\frac{p_{1,x}^o}{\hbar k} \end{bmatrix} \text{ and } \mathbf{s}_2 = \begin{bmatrix} -\frac{p_{2,z}^o}{\hbar k} \\ +\frac{p_{2,y}^o}{\hbar k} \\ +\frac{p_{2,x}^o}{\hbar k} \end{bmatrix} \tag{12}$$

Equations (11) and (12) reveal that there is strong, *transverse*, spin-orbit coupling for both left-propagating eigenwaves in the considered *isotropic* inhomogeneous (i.e., not anisotropic homogeneous, as usually) medium. Crucially, they also reveal, from a spin-orbit interactions perspective[11] too, a further reason for the attained one-way property. For a left-going ($p_x < 0$) wave incident at an arbitrary angle on the $x = 0$ plane, where there is continuity of the momentum components $p_y$ and $p_z$, together with the conservation of the spin, there are two possibilities: Either the eigenwave is reflected to the same eigenwave, which is impossible because the change in the sign of $p_x$ (from $p_x < 0$ to $p_x > 0$) and the continuity of $p_y$ and $p_z$, would imply, from Eq. (12), that the spin components $s_x$ and $s_y$ of the incident and reflected waves would be equal, but their $s_z$ component would change sign – which is not allowed, owing to the conservation of the spin. The second possibility is that the first eigenwave is reflected to the second eigenwave of Eq. (8), which has a Weyl-like structure, connected at the inception uniquely to the one-way operators, but this too is excluded because with $p_x < 0$, $p_y > 0$ and $p_z > 0$ for the first (incident) eigenwave and $p_x > 0$, $p_y > 0$, $p_z > 0$ for the second (reflected) eigenwave, we see from Eq. (12) that this implies $s_x > 0$, $s_y < 0$, $s_z > 0$ for the first eigenwave and $s_x < 0$, $s_y > 0$, $s_z > 0$ for the second eigenwave, which is again not allowed, owing to the conservation of the spin angular momentum. Thus, for all scenarios, and for all angles of incidence, reflection from the $x = 0$ plane is rigorously suppressed – as expected in the first place from the present exact solution for the 'Engquist-Majda' operator $L^-$.

Finally, we may formally identify the topological nature[5] of the afore-described strong spin-orbit interactions. In particular, in both cases we may calculate the Berry connection $\mathbf{A}(\mathbf{k}) = i\psi \cdot \nabla_k \psi$, from where we find that in both cases the Berry curvature $\boldsymbol{\Omega}(\mathbf{k}) = \nabla_k \times \mathbf{A}(\mathbf{k}) = \frac{\mathbf{k}}{2k^3}$ and its flow through the $\mathbf{k}$-space sphere $\gamma = \int_S \boldsymbol{\Omega}(\mathbf{k})d\mathbf{S} = 2\pi$, leading to a non-zero, integer Chern number $C = \gamma/(2\pi) = 1$ – completing the proof as to the topological nature of the 3D one-way wave solution(s). Exactly analogous results can similarly be obtained for all other eigenwaves, propagating in the $+x$, $\pm y$, $\pm z$ directions.

## Discussion

As an example of the power of the above discovery, we shall now _systematically_ design and engineer a 3D one-way device using Eq. (7) and (8), for which we can be **certain** – right from the beginning, owing to the above properties of Eq. (7) – that it is a topological one. Indeed, from the 'generator' Eq. (7), let us be steered by the properties of the sigma matrices (above Eq. (7)), and, in a targeted way, modify, e.g., $k_z$ to, say, $b_0 - b_1\cos(k_z)$, where $b_0$, $b_1$ are simply two arbitrary constants. We want our 3D *one-way* material to be made of multiple layers, and for each layer we want to have, say, a 2D honeycomb lattice structure, where each unit cell has two inequivalent sites (A and B sublattices). The so-designed 3D crystal structure consists of repeating the above 2D layers periodically along the z-axis. The in-plane lattice vectors are then:

$$\mathbf{a}_1 = a\hat{x}, \; \mathbf{a}_2 = \frac{a}{2}\hat{x} + \frac{\sqrt{3}a}{2}\hat{y}, \; \mathbf{a}_3 = -\frac{a}{2}\hat{x} + \frac{\sqrt{3}a}{2}\hat{y}, \tag{13}$$

The nearest-neighbor displacement vectors connecting sub-lattices A and B are:

$$\mathbf{d}_1 = (0, -a), \ \mathbf{d}_2 = \left(\frac{\sqrt{3}a}{2}, \frac{a}{2}\right), \ \mathbf{d}_3 = \left(-\frac{\sqrt{3}a}{2}, \frac{a}{2}\right) \quad (14)$$

Starting from the real-space tight-binding model with hopping amplitudes $t_{xy}$ and $t_z$ the momentum-space Hamiltonian $H(\mathbf{k})$ takes the form:

$$H(\mathbf{k}) = \begin{pmatrix} b_0 - b_1 \cos(k_z a_z) & t_{xy}(1 + e^{i\mathbf{k}\cdot\mathbf{a}_1} + e^{i\mathbf{k}\cdot\mathbf{a}_2}) \\ t_{xy}(e^{-ik_x a} + e^{-i\frac{k_x a}{2} + i\frac{\sqrt{3}a}{2}k_y} + e^{-i\frac{\sqrt{3}a}{2}k_y}) & -(b_0 - b_1 \cos(k_z a_z)) \end{pmatrix} \quad (15)$$

From this, we may analytically find the Berry curvature near a band inversion point:

$$\Omega_{xy}(k_x, k_y, k_z) = \frac{b_0 - b_1 \cos(k_z)}{2(k_x^2 + k_y^2 + (b_0 - b_1 \cos(k_z))^2)^{3/2}} \quad (16)$$

Finally, the analytically derived Chern number is given by:

$$C(k_z) = \begin{cases} 1, \text{ if } |b_0 - b_1 \cos(k_z a_z)| < 0 \text{(band inversion)} \\ 0, \qquad\qquad\qquad\qquad\qquad\qquad \text{otherwise} \end{cases} \quad (17)$$

which is precisely what we were looking for.

In conclusion, we have identified the exact, 3D one-way wave equation, starting from Engquist and Majda's seminal work[5], but assuming spinor eigenfields[5,10,11]. The discovered equation(s), under judicious excitations give rise to solely one-way wave propagation, and, surprisingly, turn out to have a deeply topological nature, a feature that could not be discerned by the approximative previous solutions[5,7-9,12,13]. Our exact one-way wave equation(s) – not necessarily restricted to electromagnetic waves[14], but concerning **all** types of waves in many contexts – may guide <u>systematic</u> new designs of one-way devices *without any direct reference to, e.g., an interplay between topology and gapless modes, sign of the group velocity, or space-/time-symmetries*, simply by being steered by the analytical one-way <u>wave</u> equations and their properties, thereby conceivably allowing for a fundamental shift of emphasis from topology directly to the in-built nature of the *one-way wave*-propagation itself.

## Data availability
The data that support the findings of this study are available from the corresponding author upon request.

## Code availability
The simulation codes used in this study are available from the corresponding author upon request.

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

## Acknowledgements
K.L.T. acknowledges support for this research by the General Secretariat for Research and Technology (GSRT) and the Hellenic Foundation for Research and Innovation (HFRI) under Grant 4509. K.L.T.'s part was also carried out within the framework of the National Recovery and Resilience Plan Greece 2.0, funded by the European Union - Next Generation EU (Implementation body: HFRI) under Grant 16909.

## Author contributions
K.L.T. conceived the idea, which was then developed technically equally by K.L.T. and T.P.S.

## Competing interests
The authors declare no competing interests.
