## [Transparent Peer Review file · Nature Communications]

Discovery of the exact 3D one-way wave equation

Corresponding Author: Professor Kosmas L Tsakmakidis

Version 0:

Reviewer comments:

Reviewer #1

(Remarks to the Author)

In this work, the authors present a method to obtain an exact one-way wave equation in three dimensions. The key idea is to borrow the treatment by Paul Dirac when dealing with the famous Dirac equation. By doing so, the authors bypass the approximation during taking the square root and obtain an exact 3D wave equation supporting one-wave propagation. Interestingly, a nonzero Chern number is also discovered as proof of the topological nature of the equation.

This is an interesting work, and the theoretical derivation looks sound. However, my main concern is how the results of this work could be useful. In other words, how can we apply the results in realistic wave systems? There are many existing wave systems supporting one-way propagation. Can they be explained by the theory here? Also, as mentioned in the conclusion part of this paper, can the authors design a realistic one-way device to demonstrate the power of the theory?

A technique issue: it is unclear what the exact implication of the Chern number is. In conventional topological phases, a nontrivial Chern number in the bulk induces one-way modes on the boundaries. Here, can the authors also prove that the Chern number defined in the paper is responsible for the one-way solution, or justify in a clear and non-suggestive way how the topological nature can be related to this Chern number?

Reviewer #2

(Remarks to the Author)

The paper is devoted to the history of a computational technique that is extremely important for modeling wave propagation. It gives an interesting, and as far as I can judge, close to exhaustive, analysis of the prehistory of the question. It seems to me methodically useful that the authors address the Dirac's factorization technique. As for the literature following the important work of Engquist and Majda, it is chosen rather selectively, however, this is inevitable given the length of the article. Authors could also mention a good review Hagstrom, T., Assaf Mar-Or, A., and Givoli, D. "High-order local absorbing conditions for the wave equation: Extensions and improvements," J. Comput. Phys. 227, 3322–3357 (2008). Also, similar approach was developed a bit later for the "parabolic equation of the diffraction theory" which is itself a kind of one-way approximation. I will mention only one work in this direction: Baskakov, V., and Popov A. "Implementation of transparent boundaries for numerical solution of the Schrödinger equation," Wave Motion, 14 123-128, (1991).

I recommend a minor revision.

Version 1:

Reviewer comments:

Reviewer #1

(Remarks to the Author)

I thank the authors for their efforts to address my questions. I now recommend publication.

We would like to sincerely thank both referees for recognizing the conceivable importance of our present work, and for raising interesting points, the addressing of which we believe has considerably, now, improved the clarity and potential impact of our work. Below, we provide a point-by-point reply to all points raised by both referees.

Reviewer #1:

In this work, the authors present a method to obtain an exact one-way wave equation in three dimensions. The key idea is to borrow the treatment by Paul Dirac when dealing with the famous Dirac equation. By doing so, the authors bypass the approximation during taking the square root and obtain an exact 3D wave equation supporting one-wave propagation. Interestingly, a nonzero Chern number is also discovered as proof of the topological nature of the equation. This is an interesting work, and the theoretical derivation looks sound.

Reply: We sincerely thank the referee for the positive appraisal of our work.

However, my main concern is how the results of this work could be useful. In other words, how can we apply the results in realistic wave systems? There are many existing wave systems supporting one-way propagation. Can they be explained by the theory here? Also, as mentioned in the conclusion part of this paper, can the authors design a realistic one-way device to demonstrate the power of the theory?

Reply: We thank the referee for this very constructive comment. Indeed, our method is based entirely on the one-way wave equation – not on a derivation of a band diagram, and then investigating the topological properties of each band. This means that we can now design – using this one-way wave equation – practical one-way (unidirectional) devices by following the terms of this equation. This property, not only leads to new topological devices, but also we completely avoid deriving any band-diagram – i.e., we are certain from the beginning that building a device by interpreting each term will necessarily lead to a one-way (unidirectional) device. In our revised manuscript, we have now designed just such a 3D one-way device, about which we knew from the beginning that it will be unidirectional, with precisely the properties needed to lead to a unity ($C = 1$) Chern number. This systematic way (from Eq. (8) of the main manuscript) of designing 3D one-way devices has never been reported before, and it is a unique feature of the present work.

A technique issue: it is unclear what the exact implication of the Chern number is. In conventional topological phases, a nontrivial Chern number in the bulk induces one-way modes on the boundaries. Here, can the authors also prove that the Chern number defined in the paper is responsible for the one-way solution, or justify in a clear and non-suggestive way how the topological nature can be related to this Chern number?

Reply: Thank you for requesting this clarification. In the revised manuscript, we now explain that our exact one-way equation turns out to have a topological nature, where ‘topological’ is meant in the standard way. An integer Chern number derived from our exact one-way wave equation, here too implies the existence of one surface wave

propagating on the interface of a material (derived from our equation) and air. The important novelty lies of course in the fact that, whereas until now we knew that topological waves are usually unidirectional, we are now led to the fundamental general insight that the opposite too is true, namely that *any* wave propagating rigorously *in a one-way manner for all angles*, must inherently be topological in nature.

Reviewer #2 (Remarks to the Author):

The paper is devoted to the history of a computational technique that is extremely important for modeling wave propagation. It gives an interesting, and as far as I can judge, close to exhaustive, analysis of the prehistory of the question. It seems to me methodically useful that the authors address the Dirac's factorization technique. As for the literature following the important work of Engquist and Majda, it is chosen rather selectively, however, this is inevitable given the length of the article.

Reply: We sincerely thank the referee for the positive appraisal of our work.

Authors could also mention a good review Hagstrom, T., Assaf Mar-Or, A., and Givoli, D. "High-order local absorbing conditions for the wave equation: Extensions and improvements," J. Comput. Phys. 227, 3322–3357 (2008). Also, similar approach was developed a bit later for the "parabolic equation of the diffraction theory" which is itself a kind of one-way approximation. I will mention only one work in this direction: Baskakov, V., and Popov A. "Implementation of transparent boundaries for numerical solution of the Schrödinger equation," Wave Motion, 14 123-128, (1991).

Reply: We thank the referee for bringing to our attention these two interesting papers, highlighting the approximative nature of all previous solution-attempts.